# Quality and Nutritional Changes of Traditional Cupcakes in the Processing and Storage as a Result of Sunflower Oil Replacements with Refined Olive Pomace Oil

**DOI:** 10.3390/foods12112125

**Published:** 2023-05-24

**Authors:** Joaquín Velasco, Aída García-González, Rosario Zamora, Francisco J. Hidalgo, María-Victoria Ruiz-Méndez

**Affiliations:** Instituto de la Grasa, Consejo Superior de Investigaciones Científicas (CSIC) Carretera de Utrera, km 1, 41013 Sevilla, Spain; aida.garcia@csic.es (A.G.-G.); rzamora@ig.csic.es (R.Z.); fhidalgo@ig.csic.es (F.J.H.); mvruiz@ig.csic.es (M.-V.R.-M.)

**Keywords:** olive pomace oil, sunflower oil, cupcakes, beating, baking, oxidation, bioactive oil components, storage, quality

## Abstract

Recent nutritional studies have shown that the regular consumption of olive pomace oil (OPO) contributes to cardiovascular and cardiometabolic disease prevention. OPO could be a healthier alternative to the polyunsaturated oils employed in a number of bakery foods. However, little is known about the quality and nutritional changes of OPO in these products, especially the amounts of its bioactive components that finally reach consumers. The aim of this research was to evaluate refined OPO as a substitute for sunflower oil (SO) in cupcakes specially manufactured with a 6-month shelf-life. The influence of processing and storage on lipid oxidative changes and the levels of OPO bioactive components was studied. OPO samples exhibited much higher resistance to oxidative degradation in the processing and especially after storage, which had a greater oxidative impact. OPO reduced considerably the levels of oxidised lipids. HPLC analysis showed hydroperoxide triglyceride concentrations of 0.25 (±0.03) mmol/kg fat against 10.90 (±0.7) mmol/kg in the control containing SO. Sterols, triterpenic alcohols and triterpenic acids remained unchanged, and only slight losses of squalene (8 wt%) and α-tocopherol (13 wt%) were observed in OPO after processing and storage, respectively. Therefore, OPO preserved its nutritional properties and improved the quality and nutritional value of the cupcakes.

## 1. Introduction

Olive pomace oil (OPO) is the oil obtained from the olive paste generated in the extraction process of virgin olive oil. The remaining oil is generally obtained by solvent extraction with hexane, refined and headed with a small amount of virgin olive oil that provides it with characteristic flavours [1]. OPO is a monounsaturated oil that meets the fatty acid composition recommended in nutritional guidelines [2,3], and it contains characteristic minor components with biological activities [4]. These components include squalene, tocopherols, triterpenic alcohols (erythrodiol and uvaol), triterpenic acids (mainly oleanolic acid), and aliphatic alcohols. A few of these components such as squalene are present in much lower amounts in other common vegetable oils or they are not found at all, as they are unique to OPO and other oils obtained from olives, such as triterpenic compounds and aliphatic alcohols [5]. Recently, the beneficial effects of long-term OPO consumption have been observed in healthy consumers and subjects at cardiovascular risk, such as hypolipidemic effects, reduction in total and LDL cholesterol, and reduction in waist circumference. These benefits were not found when sunflower oil (SO) or high oleic sunflower oil was used instead. It was concluded that the regular consumption of OPO contributes to cardiovascular and cardiometabolic disease prevention [6,7,8].

In addition to its beneficial properties, OPO has shown a high resistance to oxidative degradation in processes such as the frying of foods carried out at high temperatures [9]. A few of its bioactive components, including sterols, oleanolic acid, erythrodiol and particularly aliphatic alcohols have shown high thermal stability in the frying of different foods [10].

Even though the thermoxidative stability of OPO has been recently investigated at frying conditions [9,10], very little is known about its oxidative changes when used as an ingredient in bakery foods. While olive pomace, defatted or not, has been employed as a partial flour substitute in a variety of baked products, including biscuits [11,12], bread [13] and breadsticks [14], only a bibliographic precedent of the use of OPO in baked foods has been found. This is the work by Caponio and co-workers [15], who studied OPO as a reference refined oil to evaluate the potential use of extra virgin olive oil (EVOO) as an ingredient of “taralli”, which is a typical Italian savoury product rich in fat. “Taralli” containing OPO exhibited lower oxidative stability than EVOO during the making process and storage, but this was still relatively high and comparable to those of other two refined oils of recognised high oxidative stability such as olive oil and refined palm oil, which were both used also as references. To the best of our knowledge, no research has been conducted so far on to what extent OPO preserves its bioactive components during the processing and storage of baked foods.

Because of its high contents of oleic and low levels of linoleic acid, it is hypothesised that OPO resists better the processing and storage conditions of bakery foods than polyunsaturated oils, such as sunflower oil (SO), which is normally employed in the preparation of a number of these products. SO is considerably susceptible to oxidative degradation due to its elevated content of linoleic acid [16], which can vary between 48% and 74% of total fatty acids [17]. Lipid oxidation is one of the most important causes of quality deterioration in numerous foods. Oxidised lipids impart off-flavours that make foods unacceptable to consumers [16].

The main aim of this study was to explore the possibilities of OPO as a substitute of SO in baked foods in order to improve their nutritional value and oxidative stability. The nutritional value would be increased by a remarkable improvement in the fatty acid composition and the incorporation of the bioactive components of OPO, some of which are present in lower amounts in SO or not at all [5]. In addition, if OPO were able to reduce the levels of lipid oxidation products in such foods, it would also contribute to improving the nutritional value, as generally, oxidised lipids are considered to have potential toxicity, and consequently, their intake should be reduced [18,19].

Traditional cupcakes without any topping, neither frosted nor iced, were used in this study as a typical baked food of relatively high consumption [20]. Cupcakes belong to the category of bakery and pastry foods named alveolar baked products (ABP) that are commonly made with flour, sugar, eggs, fat or oil and leavening [21]. The manufacture of ABP has faced many challenges due to changes in consumer behaviour and eating habits [22]. Consumers demand fresh-like, pleasant taste and healthier foods. In this regard, the use of OPO in cupcakes would be a healthier alternative to SO, which is normally employed in these products.

Refined OPO was used in this study instead of the one commercialised to the public, i.e., headed with virgin olive oil, because it is a bland oil, like refined SO, free of characteristic flavours. Partial and total replacements of SO were studied in cupcakes prepared by a regular manufacturer of these products. According to the manufacturer, the shelf-life of cupcakes is normally 2 months, but the products considered in this study are specially made to have a shelf-life of 6 months. The resistance to oxidative changes of the oils during the processing and shelf-life was investigated as well as the amounts of valuable OPO bioactive components that can reach the consumer. The latter can be considered one of the most novelty aspects of this research given the lack of information available so far.

## 2. Materials and Methods

### 2.1. Oils and Cupcakes

The oils of this study were refined oils acquired from a local supplier (COREYSA, Osuna, Seville, Spain). Blends of SO and OPO containing, respectively, 25 wt% (OPO25) or 50 wt% OPO (OPO50) were prepared stepwise using the stirring tank of a 25-L FT66 Armfield oil neutraliser–washer–bleacher reactor (Armfield Ltd., Ringwood, UK). The tank featured a paddle impeller, and the blends were made gently at room temperature for 10 min. The oils were provided to a manufacturer of processed bakery and pastry goods (CODAN S.A., Arganda del Rey, Madrid, Spain) for the elaboration of cupcakes. Twelve different products containing different types of oil were produced (Table 1). The ingredients used were those regularly employed by the manufacturer, and only the oil was changed in the processing. Wheat flour, oil, sugar, liquid egg, water, whey, glycerol, leavening, sorbitol, common salt, potassium sorbate and lemon essence were employed. The weight proportions of main ingredients in the batter were 31.1% flour, 20.7% oil, 19.4% sugar, 15.9% liquid egg, and 11.1% water. The amount of batter prepared in each batch was 5.22 kg, and three batches were made with each oil. Except the flour, the solid and liquid ingredients were mixed in a beater at intermediate speed for 10 min. Then, the flour was added and beaten at maximum speed for 2 min. The batter was left to stand for 40 min. Square cupcakes from 27 g batter each were baked in an oven at 190 °C for 13 min 40 s.

Packages of 12 units with a net weight of 280 g were produced. The cupcakes were first packed in pairs using regular polypropylene films (30 μm thickness), and the pairs were in turn packed employing double films of polypropylene of 20 μm thickness each. The safety of the cupcakes was guaranteed by microbiological control carried out by the manufacturer on fresh samples and samples stored over the shelf-life, i.e., 6 months. The microbiological control consisted of the detection and/or enumeration of *Escherichia coli* by method ISO 7251:2005 [23], *Staphylococci* according to method ISO 6888-3 [24], *Salmonella* spp. applying method ISO 6579-1:2017 [25], sulphite-reducing bacteria growing under anaerobic conditions by method ISO 15213:2003 [26], microorganisms after aerobic incubation at 30 °C by method ISO 4833-1:2013 [27], and yeasts and moulds applying methods ISO 21527-1:2008 [28] and ISO 21527-2:2008 [29].

### 2.2. Oil Characterisation

Analyses of major and minor oil components were performed for the characterisation of the oils. The fatty acid composition, sterols and triterpenic alcohols, and fatty alcohols were analysed by the EU regulation [30]. Squalene was determined according to a previous report [10]. Triterpenic acids were analysed following a method reported elsewhere [31]. Tocopherols (ISO 9936:2016), acidity (ISO 660:2020) and peroxide value (PV) (ISO 3960:2017) were determined applying ISO methods [32,33,34]. The oxidative stability index (OSI) was determined at 110 °C in the Rancimat test applying AOCS Cd 12b-92 method [35]. Triacylglycerol polymers were measured following ISO method 16931:2010 [36]. Triglycerides bearing linoleate hydroperoxides or hydroperoxydienes were quantitatively analysed by HPLC-UV following a method developed in our lab [37].

### 2.3. Cupcake Characterisation

In order to examine whether the type of oil had an impact on the physicochemical properties of the cupcakes, the moisture content, the amount of oil and the colour were evaluated.

The moisture content was determined in an Ohaus MB45 analyser with a halogen heating source (Ohaus Corporation, Parsippany, NJ, USA). The fat content was obtained by Söxhlet extraction with hexane according to ISO method 734:2015 [38]. In both cases, the samples were crumbled with gloved hands.

Colourimetric measurements on cupcakes were performed with a PCE-CSM8 colour-view spectrophotometer (PCE Instruments, Alicante, Spain) equipped with software to calculate the CIE L* (lightness), a* (redness) and b* (yellowness) parameters. Measurements were taken on three points of a cupcake, and 56 units were considered for each sample. The colour index (CI) was determined according to the formula CI = L*(b* − a*)/100 reported elsewhere [39].

The OSI was also determined in intact cupcake samples to obtain in a quick way an insight on the expected improvement in the oxidative stability provided by the SO replacements with OPO. The Rancimat test applied at 110 °C to 5 g crumbled cupcake samples was used for the OSI determination.

### 2.4. Fat Extraction

In order to obtain the fat extract in the batter, the product was previously freeze-dried in a LyoQuest lyophiliser (Telstar S.A., Madrid, Spain) and ground in a mortar and pestle. The amount of water removed was 22.9 ± 0.5%. The fat extract was obtained from the ground product (22 g) with n-hexane (100 mL) applying stirring for 10 min at room temperature in inert headspace (N_2_) in the dark. The product was washed twice with n-hexane (50 mL each) to extract the remaining fat, and the solvent was removed in a rotary evaporator at 30 °C and then with a stream of N_2_. The fat extracts in the cupcakes were obtained applying the same procedure, but the moisture was not removed prior to extraction. The product was previously crumbled with gloved hands.

### 2.5. Oxidative Changes in the Processing

Oxidative changes in the beating and baking processes were evaluated by applying the PV [34], the levels of triglyceride hydroperoxydienes [37], the levels of polymers [36], as well as the contents of polar compounds [40] in the baking step. The method of polar compounds was based on the combination of ISO methods 8420:2002 [41] and 16931:2010 [36]. ATR-FTIR spectroscopy analysis was applied to confirm the absence of hydroperoxides at substantial amounts in the batter samples. A Bruker 55 Equinox S FTIR spectrometer with a DGTS detector (Bruker Optics, Ettlingen, Germany) and a detachable attenuated total reflectance (ATR, six bounces, Specac, Orpington, UK) cell consisting of a zinc selenide crystal was used. The spectra were recorded at room temperature in the region of 4000–600 cm^−1^ at a resolution of 4 cm^−1^. A volume of 250 μL was spread uniformly through the ATR crystal and analysed in triplicate. Spectra were processed with OPUS programme version 4.0 (Bruker Optics, Ettlingen, Germany).

The amounts of bioactive components, tocopherol, squalene, sterols and triterpenic alcohols, and triterpenic acids, were also measured as outlined above to evaluate possible losses during baking.

### 2.6. Storage Assay

Randomly selected samples of cupcakes, namely, SO-1, OPO25-1, OPO50-1 and OPO-1, packed in original packages of polypropylene containing 12 units, were stored at 23 ± 2 °C in the dark and analysed at 2-month intervals over the shelf-life period, i.e., 6 months according to the manufacturer. The minimum and maximum temperatures were, respectively, 16.5 °C and 26.5 °C.

Oxidative changes were evaluated by determining the levels of hydroperoxydienes in the fat extracts according to a previous report [37]. Analysis of volatiles by SPME-GC-MS in samples crumbled by gloved hands was applied to evaluate the levels of hexanal as a marker of secondary oxidation (Method S1).

The cupcakes were also assessed by sensory analysis. The analyses were performed by three expert panellists of the official panel of the Instituto de la Grasa (CSIC) who have been trained for the sensory assessment of virgin olive oils. They only evaluated rancidity in an open session on cupcakes taken from a given package. They first smelled a sample and then tested it to assess the aroma and flavour, respectively. The perception intensity was measured on a structured 9-point scale in which 0 was given when rancidity was not perceptible, 2 for slightly perceptible, 4 for perceptible, 6 for considerably perceptible, 8 for strongly perceptible and 9 for very strongly perceptible. The panellists were periodically trained using sunflower oil oxidised at 60 °C for 1 week and its dilutions with fresh oil.

The amounts of bioactive oil components were also measured as outlined above to evaluate possible losses during storage.

### 2.7. Consumer Tests

The cupcake samples SO-1, OPO25-1, OPO50-1 and OPO-1 were tested by a panel of 61 regular consumers of cupcakes (33 women and 28 men, 30% age 18–31, 39% age 32–45 and 31% age 46–60) to determine the acceptability and consumer preferences for the products containing OPO. Fresh samples and cupcakes stored for 3 and 6 months were evaluated. The manufacturer of cupcakes guaranteed the safety of the fresh and stored samples by microbiological control. The assays were carried out by AINIA Consumer (Paterna, Spain), which was a company specialised in consumer tests. The consumers were served a unit of each product on plates randomly coded with three digits. A sequential monadic serving order was employed. The attributes assessed were global appraisal, appearance, fresh appearance, colour, aroma, flavour, taste intensity, sweet taste, texture, sponginess, hard–soft texture, dry–juicy texture and fresh in mouth. A structured 9-level hedonic scale in which level 1 stands for “I do not like it at all” and level 9 stands for “I like it very much” was employed. Generally, high scores concerned positive aspects. For instance, level 1 in the attribute hard–soft texture denoted very hard, whereas level 9 indicated very soft. The “taste intensity” and “sweet taste” were evaluated using a structured 5-level hedonic scale in which level 1 stands for “much milder than I would like” or “much less sweet than I would like”, level 2 stands for “a little milder or a little less sweet than I would like”, level 3 stands for “just as I like”, level 4 stands for “a little more intense or a little sweeter than I would like” and level 5 stands for “much more intense or much sweeter than I would like”. Following instructions, the consumers first observed, smelled and tasted a sample and evaluated global appraisal, appearance, colour and aroma. Then, they tasted it again to evaluate the rest of the attributes. They were also asked to indicate whether they perceived the rancid flavour of seed oils or nuts. Finally, the consumers were served the 4 types of cupcakes at a time to rank them in order of preference, considering the first and fourth positions for the best and worst assessed, respectively.

Based upon a database of sensory acceptance of over 4000 tested foods, the consumer test company classifies products as a function of sensory acceptance scores on a 9-level hedonic scale in the following categories: very well assessed (>6.7), well assessed (6.1–6.7), moderately assessed (5.6–6.0), badly assessed (3.5–5.5) and very badly assessed (<3.5).

### 2.8. Statistical Analyses

The fresh oils and batter fat extracts were analysed in triplicate. Unless indicated, the cupcakes were also analysed in triplicate, taking three cakes from a single package and analysing each only once. Six fresh oils, i.e., three SOs and three OPOs, and 12 batter fat extracts, i.e., one for each fresh cupcake sample, were analysed. A total of 36 cupcake extracts from fresh samples and 12 cupcake extracts from samples stored in each time interval were analysed. Results were expressed as means ± standard deviations. One-factor analysis of variance was applied. Levene’s test based on the mean value was used to verify that variances were equal across groups. For multiple mean comparisons, the Duncan test or Kruskal–Wallis non-parametric test were used. Comparisons between two samples were made by Student’s *t*-test. Friedman’s non-parametric test was applied to determine significance in the differences obtained in the consumer preference tests. Significance was defined at *p* < 0.05. The analyses were performed using the 29.0 IBM SPSS Statistics program (IBM Corp., Chicago, IL, USA).

## 3. Results and Discussion

### 3.1. Oil Characterisation

The oils of this study presented, respectively, fatty acid compositions of high-linoleic SOs [17] and OPOs [1] (Appendix A). The fatty acid compositions of the oil blends were those expected from the oil proportions employed (Appendix A). The bioactive components evaluated in the oils presented levels within reported values (Table 2) [5,9]. With regard to triterpenic acids, their levels in the OPOs were low (59–64 mg/kg) and characteristic of refined oils [5]. While the three OPOs presented no significant differences in the total content of sterols (*p* = 0.437) or triterpenic acids (*p* = 0.842), they differed in the levels of triterpenic alcohols, α-tocopherol, squalene and especially fatty alcohols (Table 2).

The acidity of the oils (0.07–0.11%) was indicative of high-quality refined oils (Appendix A). With the exception of SO3, the peroxide value (PV) was within normal ranges for fresh refined oils [1,17]. The PV of SO3 was just above the limit for fresh refined seed oils (<10 meq/kg). This oil was deliberately chosen to consider a greater level of quality variability among the oils. The OSI values of the SOs (4.62–5.65 h at 110 °C) and OPOs (14.9–18.0 h at 110 °C) were within normal ranges (Appendix A) [9]. As expected, the oil blends presented intermediate values of quality parameters (Appendix A).

### 3.2. Characterisation of the Fresh Cupcakes

The cupcakes presented moisture contents of 14.0% (±0.9) and fat content of 24.8% (±0.7) (Table 3). Statistical analysis applying ANOVA did not show significant differences in the moisture (*p* = 0.357) or fat content (*p* = 0.325) between samples prepared with different types of oil. According to Kruskal–Wallis’ test, no significant differences (*p* = 0.229) were found for the colour index either (Table 3). Therefore, the oil substitution did not cause any significant changes in these cupcake parameters.

The OSI values of intact samples of cupcakes were high (Table 3) compared to those of the oils used in their preparation (Appendix A). This fact may be related to the relatively high moisture content in the intact cupcakes, which can have a stabilising effect against lipid oxidation, and the possible formation of antioxidants during the baking process as a consequence of Maillard reactions [16]. As expected, clear differences were found in the OSI values between samples prepared with different types of oil. Thus, the OPO samples were the most stable (average value of 48.9 h), which were followed by OPO50 (average value of 20.8 h), OPO25 (average value of 15.3 h) and SO (average value of 11.5 h).

### 3.3. Oxidative Deterioration and Losses of Oil Bioactive Components in the Processing

The evaluation of the oil oxidative deterioration during the beating step was first approached using the PV. However, the fat extracts from the batter samples showed anomalous results that were unexpectedly high (Appendix A). The presence of hydroperoxides at such high amounts in the fat extracts was ruled out by applying ATR-FTIR analysis. The absorption band assigned to the O-H bond of the hydroperoxide group in the fat extracts, whose maximum is recorded at 3430 cm^−1^ [42], was not substantially different from that in the oils (Appendix A). Therefore, the overrated PV in the extracts may have been related to the coextraction of batter components, different from lipids, interfering in the analysis by participating in the iodide oxidation on which the redox titration is based.

Oxidative changes in the beating step were also evaluated by the direct HPLC analysis of triglycerides bearing linoleate hydroperoxides [37]. This analytical approach has proven to be useful for evaluating the oxidation extent in oils containing linoleate and oleate as the only oxidisible substrates present in significant amounts. The oxidation of oleate seems to be efficiently inhibited by the presence of tocopherol. Even in monounsaturated oils, such as high oleic sunflower oil, in which the amount of linoleate is relatively very low, the oxidation of oleate was not substantial until the PV was higher than 50 meq/kg [43]. In the present study, the analysis of hydroperoxide triglycerides or hydroperoxydienes did provide coherent results that showed significant oxidation in the beating (Figure 1). Oxidation in the beating step of sponge cakes has been reported, and it was mainly attributed to enzymatic oxidation caused by the lipoxygenase activity of the flour [44,45]. The oxidation levels found in the present study were low in the OPO samples and relatively high in the SO, which can be attributed to the differences in the linoleic acid content.

Hydrolytic degradation of the fat material during the beating was not directly evaluated in this study. However, the direct analysis by HPSEC of the fat extracts employed to determine the content of triglyceride polymers provided some information in this regard. Results obtained from the SO and OPO samples showed significant slight increments of monoglycerides and the group of compounds comprising the free fatty acids, but no significant increase was observed for diglycerides (Appendix A). Therefore, such increments may be mainly related to the egg fat contribution to the lipid extract rather than lipid hydrolysis during the process, as similar increments of diglycerides would have also been observed. The fact that no significant differences in the contents of diglycerides were found between the extracts and the oils suggests that lipid hydrolysis was negligible during the beating or that the HPSEC analysis was not sensitive enough to detect very slight hydrolytic changes.

In order to evaluate the thermal impact of the baking process, the levels of polar compounds in the batter and cupcakes of the samples containing SO and OPO were first analysed. No thermoxidative degradation was found in the baking step applying this determination (Table 4). Polymerisation compounds, i.e., triglyceride dimers (TGD), remained unchanged, which supported the results found in sponge cakes by other authors [46]. Similarly, oxidised triglyceride monomers (oxTGM) did not vary significantly either. In contrast, a significant slight rise was observed for the group of compounds comprising the free fatty acids. Nevertheless, such a rise cannot be attributed to the hydrolysis of triglycerides, as diglycerides and monoglycerides did not change significantly.

The direct HPLC analysis of hydroperoxydiene triglycerides, i.e., linoleate hydroperoxides, was also utilised to evaluate oxidative changes in the baking step. Unlike the analysis of polar compounds, the levels of hydroperoxydienes did exhibit significant oxidation (Figure 2). Oxidative changes during the baking were low and below the sensitivity of the analysis of polar compounds. The levels of hydroperoxides were low in the OPO samples and relatively high in the SO samples, whereas the oil blends presented intermediate levels. Similar to the beating step, the OPOs showed the largest resistance to oxidative changes during the baking process. The oxidative degradation found in the baking was comparable to or even lower than that detected in the beating step. These results were consistent with those reported for the analysis of conjugated dienes in sponge cakes [47]. The authors observed a strong increase after beating and a slight decrease after baking, which were attributed to the thermal decomposition of hydroperoxides to form volatile oxidation compounds. They concluded that oxidation occurred very early during the beating step and to a minor extent in the baking process. In this regard, the low oxidation during the thermal treatment can be justified by the high moisture content of the product. Even though the oven temperature employed in the present study was 190 °C, the thermal stress on the fat was considerably reduced by the moisture, avoiding elevated temperatures in the inner core of the cupcakes [46].

The PV was also applied to evaluate the fat extracts of cupcakes, and it also provided high results that were comparable to those found in the batter samples, supporting the presence of analytical interferences also in the fat extracts of cupcakes (Appendix A). As far as we know, no results have been reported in this regard so far.

The evaluation of possible losses of oil bioactive components during the beating step were not considered in this study due to the fact that the egg fat contribution to the lipid extracts would have made it difficult to compare the two lipid matrices, i.e., the oil and the lipid extract. In contrast, the comparison between the batter and the final product allowed the evaluation of possible changes during the baking process. The results showed that the baking did not cause substantial losses of oil bioactive components in the cupcakes, which is coherent with the low thermal stress on the fat due to the high moisture content and the low oxidative degradation detected. Only slight losses of α-tocopherol and squalene were found. While losses of 4–9 wt% α-tocopherol were detected in a few samples containing SO or the oil blends, significant changes were not observed in the samples containing OPO (Table 5). Similarly, squalene losses of 6–8% were also detected in a few cases (Table 5). Therefore, the losses of α-tocopherol and squalene were globally very small and close to the analytical error (5%).

The content of total sterols was only evaluated in the samples that were to be studied in the storage assay. In all cases, significantly higher amounts were found in the extracts of the cupcakes compared to those of their corresponding batter samples (Table 6). The sterol composition clearly showed that these differences were due to a higher cholesterol content, whose extraction was more favoured in the final product than in the batter (results not shown). In fact, when cholesterol was subtracted from the total amount of sterols, i.e., when only phytosterols were considered, no significant losses were found between the fat extracts (Table 6). The higher levels of cholesterol found in the extracts of cupcakes could explain in part the increased levels of the group of compounds comprising free fatty acids and other minor fat components found in the analysis of polar compounds (Table 4). Similarly, no significant losses of alcohols or triterpenic acids were detected in the OPO sample as a consequence of baking (Table 7).

### 3.4. Oxidative Changes and Losses of Oil Bioactive Components during the Shelf-Life

#### 3.4.1. Oxidative Degradation

The quantitative analysis of hydroperoxide triglycerides by HPLC showed lipid oxidation in the cupcakes during the shelf-life (Figure 3A). It was evident that the storage caused a larger oxidative impact on the fat material than the processing did. The level of degradation of the OPO sample was much lower than that of the sample containing SO at the end of the shelf-life (6 months). Expectedly, the samples containing the oil blends presented intermediate values, which were higher for the one containing more SO (OPO25-1), so the oxidative degradation extent clearly depended on the degree of oil unsaturation.

The levels of lipid oxidation products detected in the control cupcakes were considerably reduced with the total or partial oil replacement with OPO. This fact is of great interest from the nutritional point of view, as lipid oxidation products are generally questioned for their detrimental health effects, and nutritional recommendations indicate that their intake should be reduced [17,18].

Despite the significant levels of oxidation detected in the cupcakes at the end of the shelf-life, especially that containing SO, oxidative rancidity was not perceived by sensory analysis in any case (not shown). In fact, no significant increases in hexanal were detected over the entire storage period (Figure 3B). The hexanal presented oscillations over a threshold value detected in the initial samples, but the results did not show significant increments. Similarly, no significant differences were observed in the profile of volatile compounds between samples prepared with different types of oil or between the fresh and stored samples (Appendix A). The main components detected were those of the lemon essence, which is characterised by the predominance of D-limonene followed by β-pinene, sabinene and α-pinene [48]. A decrease in the signal intensity of these monoterpenes was observed after storage in a concomitant way to certain loss of aroma reported by the panellists (not shown). The similarities in the volatile profile between samples prepared with different oils can be attributed to the fact that refined oils were employed in this study. The use of refined OPO instead of refined OPO headed with virgin olive oil, as it is commercialised, seemed to be suitable to replace such a bland oil as SO in cupcakes.

#### 3.4.2. Contents of Oil Bioactive Components

Only slight significant losses of α-tocopherol were found during storage (Table 8). The losses were relatively lower in the OPO sample (13%) at the end of the shelf-life (6 months) compared to the rest of the samples (15–18%), which can be attributed to the lower oxidation detected (Figure 3A). α-Tocopherol is a natural antioxidant that breaks the oxidation chain by reacting with peroxyl radicals [16]. No significant losses of squalene, phytosterols, alcohols or triterpenic acids were observed either (Appendix A). These results were also coherent with those obtained for the levels of hydroperoxides and hexanal, which showed moderate or low oxidative degradation during the shelf-life. The rates of degradation of α-tocopherol and squalene have been studied in extra virgin olive oil (EVOO) after storage at room temperature [49]. The results showed that α-tocopherol was first oxidised and that squalene was protected by α-tocopherol. The content of squalene did not decrease significantly until after 6 months of storage. In a similar study by Mousavi and co-workers, different EVOOs were stored at different conditions over 36 months. The results obtained at ambient temperature showed that α-tocopherol, squalene and phytosterols were well preserved even when the oxidation level evaluated by the PV was moderate (32.85–38.62 meq O_2_ kg^−1^), but it was relatively higher compared to that detected in the present study for the cupcakes [50]. Considering that hydroperoxides mainly comprised one hydroperoxy group per triglyceride molecule, the hydroperoxides levels, expressed in a molar basis, found by Mousavi et al. were 16.42–19.31 mmol/kg oil, i.e., half the PV. Therefore, the high stabilities of α-tocopherol, squalene and phytosterols found in the cupcakes were coherent with the results found in EVOO [50]. As for triterpenic alcohols and triterpenic acids, no bibliographic precedents on their stability at room temperature in virgin olive oils have been found.

### 3.5. Acceptability and Consumer Preferences

The results for each of the attributes evaluated in the fresh samples showed no significant differences between the cupcakes containing OPO, neat or blended with SO, and the control containing SO (Table 9). The global appraisal, with values between 6.5 and 6.7, indicated that the four products were well assessed. The preference test showed that none of the products was preferred to the others, and consequently, the replacement of SO with OPO or SO-OPO blends was not perceived by the consumers (Appendix A).

No significant differences in the attributes assessed were found between samples containing OPO, neat or blended with SO, and the control during the entire shelf-life period (Appendix A). Regardless of the type of oil, the consumers reported significant losses of global appraisal, fresh appearance, aroma, flavour and especially texture, sponginess, hard–soft texture, dry–juicy texture and freshness in mouth (Appendix A). The fresh cupcakes passed from well-assessed products (6.1–6.7 global appraisal according to the benchmark database of AINIA Consumer) to moderately assessed (5.6–6.0 global appraisal) after 3 and 6 months of storage. These quality losses were the result of staling, which has been identified as the major cause of the quality deterioration of baked foods and the main reason for consumer disapproval. Through complex physicochemical mechanisms, including starch retrogradation, staling causes crumb firming, crust softening and the consequently freshness loss [51]. Results clearly showed that these phenomena were perceived by the consumers. However, when they were asked to indicate rancid flavour characteristic of seed oils or nuts, results confirmed the absence of oxidative rancidity as shown by sensory analysis and hexanal measurements. In this regard, only six out of 61 assessors reported rancid flavour for the SO-1 sample stored for 6 months, whereas four assessors indicated it for OPO-1 (Appendix A). These results do not contradict those provided in the sensory analysis by expert panellists who were trained to identify lipid oxidation. In fact, consumers even indicated rancid taste in the fresh samples. Thus, three out of 61 assessors reported it for SO-1 sample and five reported it for OPO-1 (Appendix A).

Similar to the fresh samples, the consumers did not show particular preference for any of the products after 3-month storage (Appendix A). However, although no differences in the attributes tested were observed between the samples (Appendix A), the consumers preferred OPO-1 and OPO25-1 to the control and OPO50-1 sample at the end of the shelf-life (Appendix A).

## 4. Conclusions

The results of this study have proven that the partial or total substitution of SO with refined OPO reduces significantly the levels of lipid oxidation products in cupcakes due to its greater resistance to oxidative degradation during the processing and storage of this kind of food. The reduction in lipid oxidation products has a positive impact on the quality and nutritional value of foods. Hydroperoxides eventually decompose giving rise to volatile compounds that impart off-flavours and significantly contribute to food quality losses. In addition, oxidised lipids are considered to be detrimental to health, and it is recommended to decrease their levels in foods. Furthermore, the bioactive components of OPO remain practically unchanged during the processing and storage of cupcakes. Therefore, OPO keeps its valuable nutritional properties intact, and it has the potential to improve the quality and nutritional value of cupcakes. From a sensory point of view, the use of refined OPO has also proven to be suitable because consumers did not distinguish between cupcakes containing OPO, neat or blended with SO, and the reference containing SO.

This research can be regarded as an original reference of the use of refined OPO as a food ingredient in the confectionery industry and of its capability to preserve its quality and nutritional value in the processing and storage of foods.

## Figures and Tables

**Figure 1 foods-12-02125-f001:**
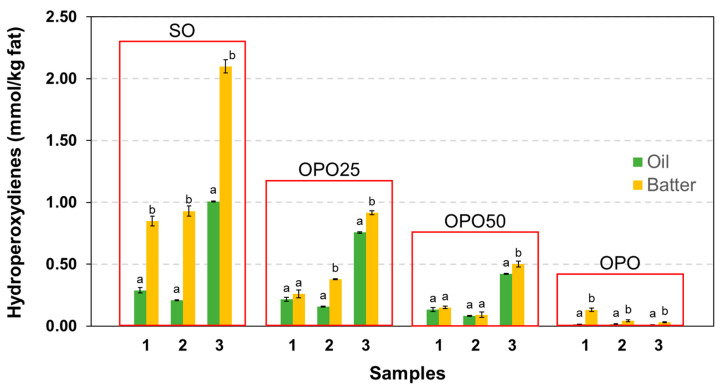
Influence of the beating process on the levels of hydroperoxydienes. Results show the mean values and standard deviation of 3 analytical determinations. Different letters indicate significant differences between the oil and batter according to Student’s *t*-test (*p* < 0.05).

**Figure 2 foods-12-02125-f002:**
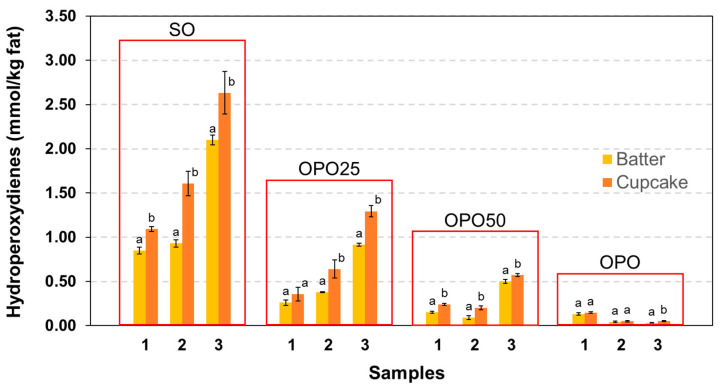
Influence of the baking process on the levels of hydroperoxydienes. Results show the mean values and standard deviation of 3 analytical determinations in an only fat extract (Batter) or in fat extracts of 3 independent samples (Cupcake). Different letters indicate significant differences according to Student’s *t*-test (*p* < 0.05).

**Figure 3 foods-12-02125-f003:**
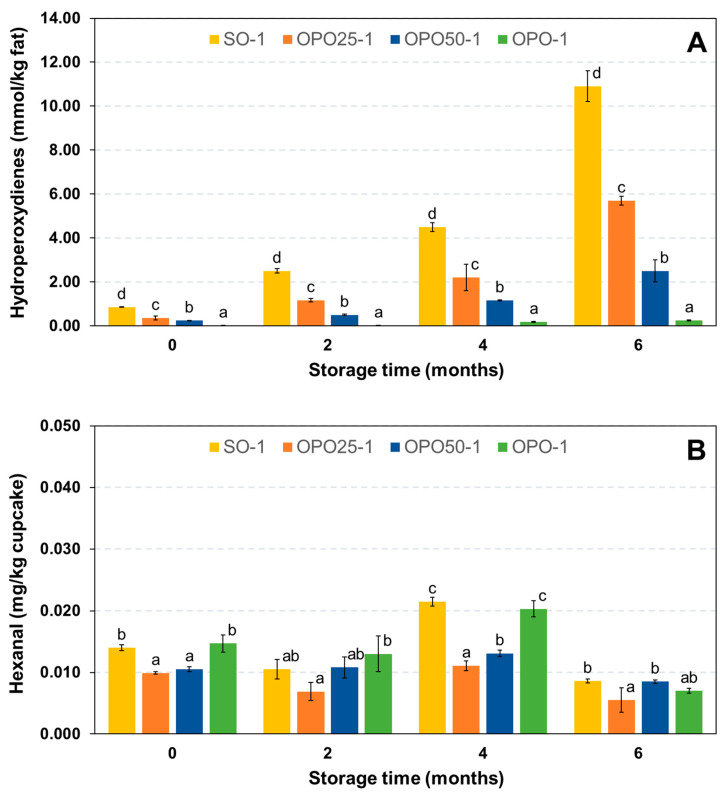
Changes in hydroperoxydienes (**A**) and hexanal (**B**) during storage. Results represent mean values and standard deviation of 3 independent samples. Different letters indicate differences between samples with the same storage time according to Duncan’s test (*p* < 0.05).

**Table 1 foods-12-02125-t001:** Oils used and manufacture date of cupcakes.

Cupcake	SO	OPO	Manufacture Date
**SO-1**	SO-1	-	29 September 2021
**SO-2**	SO-2	-	21 September 2021
**SO-3**	SO-3	-	21 September 2021
**OPO25-1**	SO-1	OPO-1	30 September 2021
**OPO25-2**	SO-2	OPO-2	21 September 2021
**OPO25-3**	SO-3	OPO-3	21 September 2021
**OPO50-1**	SO-1	OPO-1	30 September 2021
**OPO50-2**	SO-2	OPO-2	22 September 2021
**OPO50-3**	SO-3	OPO-3	22 September 2021
**OPO-1**	-	OPO-1	29 September 2021
**OPO-2**	-	OPO-2	22 September 2021
**OPO-3**	-	OPO-3	22 September 2021

SO, sunflower oil; OPO, olive pomace oil; OPO25, a blend of SO and OPO containing 25 wt% OPO; OPO50, a blend of SO and OPO containing 50 wt% OPO.

**Table 2 foods-12-02125-t002:** Contents of bioactive components in the oils.

	SO-1	SO-2	SO-3	OPO-1	OPO-2	OPO-3
Sterols (%)						
Cholesterol	0.43	0.40	0.38	0.47	0.70	0.64
	±0.11 a	±0.01 a	±0.05 a	±0.25 a	±0.17 a	±0.16 a
Brassicasterol	0.44	0.52	0.60	nd	nd	0.22
	±0.08 a	±0.09 a	±0.55 a			±0.38
24-Me-Cholesterol	0.27	0.22	0.20	nd	nd	nd
	±0.01 b	±0.02 a	±0.01 a			
Campesterol	7.85	7.90	7.95	3.13	3.03	3.25
	±0.10 a	±0.09 a	±0.40 a	±0.09 a	±0.33 a	±0.30 a
Stigmasterol	7.03	7.18	6.37	1.49	1.71	1.80
	±0.11 a	±0.16 a	±0.67 a	±0.04 a	±0.09 ab	±0.20 b
Δ7-Campesterol	2.74	2.71	2.35	nd	nd	nd
	±0.21 a	±0.14 a	±0.33 a			
Δ5,23-Stigmastadienol	0.82	0.88	0.86	0.64	0.54	0.49
	±0.01 a	±0.15 a	±0.61 a	±0.03 a	±0.47 a	±0.20 a
Clerosterol	0.13	0.14	0.16	nd	nd	nd
	±0.01 a	±0.05 a	±0.10 a			
β-Sitosterol	57.79	57.55	53.67	85.20	81.70	84.39
	±1.74 a	±0.90 a	±5.93 a	±0.43 a	±7.89 a	±1.00 a
Sitostanol	1.40	1.56	1.76	4.92	4.44	5.74
	±0.28 a	±0.12 a	±0.50 a	±0.87 ab	±0.47 a	±0.23 b
Δ5-Avenasterol	1.33	1.28	1.13	1.97	1.93	1.28
	±0.10 a	±0.07 a	±0.60 a	±0.40 a	±0.38 a	±0.38 a
Δ5,24-Stigmastadienol	3.20	3.22	2.96	2.18	1.15	1.71
	±0.29 a	±0.14 a	±0.40 a	±0.95 a	±0.29 a	±0.49 a
Δ7-Stigmastenol	13.58	13.54	13.57	nd	nd	nd
	±1.02 a	±0.56 a	±1.50 a			
Δ7-Avenasterol	2.95	2.89	3.83	nd	nd	nd
	±0.30 a	±0.13 a	±0.53 b			
**Total sterols (mg/kg)**	2451	2144	2811	2525	2495	2683
	±105 a	±155 a	±622 a	±51 a	±114 a	±284 a
**Triterpenic alcohols (%)**	-	-	-	18.5	22.7	21.7
				±0.7 a	±1.3 b	±1.1 b
**α-Tocopherol (mg/kg)**	748	732	694	334	394	412
	±7 b	±27 b	±13 a	±5 a	±3 b	±6 c
**Squalene (mg/kg)**	62	63	115	1451	1674	1439
	±3 a	±5 a	±6 b	±51 a	±67 b	±60 a
**Triterpenic acids (mg/kg)**	-	-	-	60	64	59
				±7 a	±15 a	±9 a
**Fatty alcohols (mg/kg)**	-	-	-	1733	1774	1145
				±67 b	±61 b	±111 a

nd, not detected. Results represent the mean and the standard deviation of 3 analytical determinations (*n* = 3). Different letters in a row indicate significant differences between sunflower oils (SO) or between olive pomace oils (OPO) according to Duncan’s test (*p* < 0.05).

**Table 3 foods-12-02125-t003:** Physicochemical characterisation of the cupcakes.

Sample	Moisture Content (%)	Fat Content (%)	Colour Index	OSI * (h)
**SO-1**	13.5 ± 0.9 abcA	26.1 ± 0.5 eA	8.3 ± 5.0 aA	13.0 ± 0.3 bA
**SO-2**	14.9 ± 0.6 cA	24.3 ± 0.3 abA	13.3 ± 4.6 deA	11.6 ± 0.1 abA
**SO-3**	12.4 ± 0.1 aA	24.3 ± 0.1 abA	13.3 ± 3.9 deA	10.0 ± 0.6 aA
**OPO25-1**	14.6 ± 0.3 cA	25.5 ± 0.3 deA	10.0 ± 4.2 abcA	16.7 ± 0.1 cB
**OPO25-2**	14.0 ± 0.5 bcA	24.9 ± 0.2 bcdA	13.8 ± 3.6 deA	16.1 ± 0.7 cB
**OPO25-3**	13.1 ± 0.2 abA	24.3 ± 0.4 abA	12.5 ± 3.5 cdeA	13.2 ± 0.3 bB
**OPO50-1**	14.8 ± 0.4 cA	25.1 ± 0.2 cdA	9.2 ± 4.3 abA	22.6 ± 1.3 eC
**OPO50-2**	13.8 ± 0.9 bcA	24.1 ± 0.4 aA	13.1 ± 4.5 deA	20.9 ± 1.2 eC
**OPO50-3**	13.9 ± 0.8 bcA	24.1 ± 0.5 aA	14.4 ± 2.8 eA	18.9 ± 0.4 dC
**OPO-1**	14.4 ± 1.0 bcA	25.5 ± 0.2 deA	11.6 ± 4.3 bcdA	51.8 ± 1.5 hD
**OPO-2**	14.2 ± 1.1 bcA	24.4 ± 0.6 abA	11.3 ± 5.0 bcdA	48.8 ± 3.0 gD
**OPO-3**	14.5 ± 0.9 cA	24.8 ± 0.3 bcA	11.8 ± 5.1 cdA	46.3 ± 1.1 fD

* OSI, oxidative stability index determined on intact cupcakes in the Rancimat test at 110 °C. Results represent the mean value followed by the standard deviation of 3 analytical determinations (*n* = 3) or 40 (*n* = 40) for the colour index. Different lowercase letters indicate significant differences between samples regardless of the type of oil according to Duncan’s test (*p* < 0.05), and different uppercase letters show significant differences between samples containing different types of oil according to Duncan’s test (*p* < 0.05) or Kruskal–Wallis’ test (*p* < 0.05) for the colour index.

**Table 4 foods-12-02125-t004:** Influence of the baking process on the levels of polar compounds (g/100 g fat extract) in cupcakes prepared with sunflower oil (SO) or olive pomace oil (OPO).

Sample		TGD (%)	oxTGM (%)	DG (%)	MG (%)	FFA (%)	TPC (%)
**SO-1**	Batter	0.60 ± 0.01 a	1.28 ± 0.04 a	1.28 ± 0.02 a	0.22 ± 0.01 a	0.51 ± 0.01 a	3.89 ± 0.05 a
	Cupcake	0.57 ± 0.02 a	1.28 ± 0.09 a	1.30 ± 0.03 a	0.19 ± 0.02 a	0.66 ± 0.03 b	4.00 ± 0.07 a
**SO-2**	Batter	0.56 ± 0.01 a	1.12 ± 0.02 a	1.20 ± 0.02 a	0.22 ± 0.01 a	0.41 ± 0.04 a	3.52 ± 0.04 a
	Cupcake	0.53 ± 0.04 a	1.29 ± 0.06 b	1.19 ± 0.03 a	0.20 ± 0.02 a	0.53 ± 0.01 b	3.74 ± 0.12 b
**SO-3**	Batter	0.91 ± 0.04 a	1.19 ± 0.10 b	1.03 ± 0.04 a	0.17 ± 0.01 a	0.45 ± 0.04 a	3.76 ± 0.17 a
	Cupcake	0.89 ± 0.04 a	0.99 ± 0.06 a	1.07 ± 0.03 a	0.17 ± 0.01 a	0.62 ± 0.02 b	3.73 ± 0.11 a
**OPO-1**	Batter	1.05 ± 0.05 a	1.41 ± 0.05 a	5.99 ± 0.26 a	0.66 ± 0.02 b	0.84 ± 0.04 a	9.94 ± 0.28 a
	Cupcake	1.05 ± 0.08 a	1.59 ± 0.13 a	5.72 ± 0.23 a	0.57 ± 0.02 a	1.02 ± 0.05 b	9.94 ± 0.44 a
**OPO-2**	Batter	0.91 ± 0.04 a	1.41 ± 0.09 a	6.77 ± 0.42 a	0.71 ± 0.02 a	0.86 ± 0.10 a	10.66 ± 0.55 a
	Cupcake	0.86 ± 0.03 a	1.32 ± 0.07 a	6.45 ± 0.16 a	0.69 ± 0.01 a	1.13 ± 0.02 b	10.44 ± 0.16 a
**OPO-3**	Batter	1.01 ± 0.13 a	1.18 ± 0.18 a	6.80 ± 0.09 a	0.68 ± 0.01 b	0.81 ± 0.02 a	10.48 ± 0.17 a
	Cupcake	0.94 ± 0.02 a	1.24 ± 0.03 a	6.73 ± 0.07 a	0.62 ± 0.01 a	1.12 ± 0.03 b	10.65 ± 0.02 a

TGD, triglyceride dimers; oxTGM, oxidised triglyceride monomers; DG, diglycerides; MG, monoglycerides; FFA, free fatty acids and other polar minor fat components; TPC, total polar compounds. Results represent the mean value followed by the standard deviation of 3 analytical determinations in an only oil extract (Batter) or in oil extracts of 3 independent samples (Cupcake). Different letters for a given group of compounds indicate significant differences between batter and cupcake according to Student’s *t*-test (*p* < 0.05).

**Table 5 foods-12-02125-t005:** Influence of the baking process on the contents of α-tocopherol and squalene.

Sample	α-Tocopherol (mg/kg Fat)		Squalene (mg/kg Fat)	
	Batter	Cupcake	Batter	Cupcake
**SO-1**	792 ± 5 a	762 ± 9 b	65 ± 8 a	76 ± 11 a
**SO-2**	732 ± 32 a	665 ± 25 b	63 ± 1 a	63 ± 1 a
**SO-3**	653 ± 30 a	646 ± 15 a	102 ± 5 a	107 ± 2 a
**OPO25-1**	727 ± 22 a	678 ± 12 b	391 ± 18 a	359 ± 12 a
**OPO25-2**	636 ± 21 a	665 ± 3 a	386 ± 20 a	367 ± 18 a
**OPO25-3**	612 ± 17 a	633 ± 6 a	367 ± 6 a	344 ± 9 b
**OPO50-1**	607 ± 8 a	572 ± 19 b	678 ± 15 a	621 ± 8 b
**OPO50-2**	567 ± 41 a	608 ± 4 a	665 ± 11 a	610 ± 7 b
**OPO50-3**	569 ± 17 a	578 ± 22 a	587 ± 23 a	555 ± 9 a
**OPO-1**	392 ± 4 a	377 ± 15 a	1404 ± 23 a	1286 ± 52 b
**OPO-2**	417 ± 15 a	449 ± 26 a	1374 ± 42 a	1304 ± 27 a
**OPO-3**	419 ± 4 b	435 ± 6 a	1237 ± 10 a	1121 ± 89 a

Results represent the mean value followed by the standard deviation of 3 analytical determinations in an only fat extract (Batter) or in fat extracts of 3 independent samples (Cupcake). Different letters indicate significant differences between batter and cake according to Student’s *t*-test (*p* < 0.05).

**Table 6 foods-12-02125-t006:** Influence of the baking process on the total content of sterols and phytosterols.

Sample	Total Sterols (mg/kg Fat)		Total Phytosterols (mg/kg Fat)	
	Batter	Cupcake	Batter	Cupcake
**SO-1**	2969 ± 124 a	4207 ± 173 b	2598 ± 110 a	2346 ± 164 a
**OPO25-1**	3361 ± 159 a	4872 ± 125 b	2833 ± 211 a	2628 ± 155 a
**OPO50-1**	3085 ± 203 a	4029 ± 221 b	2692 ± 212 a	2370 ± 154 a
**OPO-1**	3202 ± 63 a	4219 ± 106 b	2750 ± 206 a	2448 ± 121 a

Results represent the mean value followed by the standard deviation of 3 analytical determinations in an only fat extract (Batter) or in fat extracts of 3 independent samples (Cupcake). Different letters indicate significant differences between batter and cupcake according to Student’s *t*-test (*p* < 0.05).

**Table 7 foods-12-02125-t007:** Influence of the baking process on the content of triterpenic compounds (mg/kg fat extract) in sample OPO-1.

	Batter	Cupcake
**Triterpenic alcohols**	607 ± 19 a	610 ± 21 a
Erythrodiol	521 ± 16 a	532 ± 17 a
Uvaol	86 ± 3 a	78 ± 6 a
**Triterpenic acids**	59 ± 12 a	66 ± 9 a
Oleanolic	49 ± 8 a	54 ± 6 a
Ursolic	10 ± 4 a	12 ± 3 a
Maslinic	nd	nd

nd, not detected. Results represent the mean value followed by the standard deviation of 3 analytical determinations in an only fat extract (Batter) or in fat extracts of 3 independent samples (Cupcake). Different letters in a row indicate significant differences according to student’s *t*-test (*p* < 0.05).

**Table 8 foods-12-02125-t008:** Contents of α-tocopherol (mg/kg fat) over the shelf life.

Sample	Time (Months)			
	0	2	4	6
**SO-1**	762 ± 9 a	716 ± 28 b	674 ± 6 c	638 ± 6 d
**OPO25-1**	678 ± 12 a	616 ± 4 b	584 ± 5 c	580 ± 5 c
**OPO50-1**	572 ± 19 a	523 ± 5 b	499 ± 3 c	468 ± 3 d
**OPO-1**	377 ± 15 a	341 ± 6 b	339 ± 3 b	328 ± 2 b

Results represent the mean value followed by the standard deviation of 3 independent samples (Cupcake). Different letters in a row indicate significant differences according to Duncan’s test (*p* < 0.05).

**Table 9 foods-12-02125-t009:** Consumer panel results for different attributes assessed in fresh cupcakes.

Attributes	Samples				ANOVA/ *p*-Value
	SO-1	OPO25-1	OPO50-1	OPO-1	
**Global appraisal**	6.5 ± 1.5 a	6.7 ± 1.3 a	6.5 ± 1.3 a	6.6 ± 1.4 a	0.813
**Appearance**	6.9 ± 1.4 a	7.0 ± 1.3 a	6.9 ± 1.5 a	6.7 ± 1.4 a	0.817
**Fresh appearance**	6.6 ± 1.6 a	6.7 ± 1.3 a	6.7 ± 1.3 a	6.7 ± 1.3 a	0.995
**Colour**	6.7 ± 1.6 a	6.9 ± 1.5 a	7.0 ± 1.4 a	7.0 ± 1.4 a	0.656
**Aroma**	6.7 ± 1.7 a	6.7 ± 1.5 a	6.7 ± 1.7 a	7.0 ± 1.4 a	0.800
**Flavour**	6.3 ± 1.6 a	6.9 ± 1.3 a	6.8 ± 1.4 a	6.8 ± 1.2 a	0.180
**Taste intensity ***	2.7 ± 0.7 a	2.8 ± 0.7 a	2.7 ± 0.7 a	2.9 ± 0.7 a	0.214
**Sweet taste ***	2.9 ± 0.7 a	3.0 ± 0.6 a	3.0 ± 0.6 a	3.1 ± 0.7 a	0.550
**Texture**	5.8 ± 1.6 a	6.0 ± 1.6 a	5.8 ± 1.5 a	6.0 ± 1.7 a	0.699
**Sponginess**	5.5 ± 1.9 a	5.8 ± 1.6 a	5.4 ± 1.7 a	5.8 ± 2.0 a	0.564
**Hard–soft texture**	5.3 ± 1.6 a	5.7 ± 1.5 ab	5.5 ± 1.4 ab	6.0 ± 1.6 b	0.093
**Dry–juicy texture**	5.1 ± 1.6 a	5.6 ± 1.7 a	5.2 ± 1.5 a	5.5 ± 1.8 a	0.332
**Freshness in mouth**	5.7 ± 1.8 a	5.9 ± 1.4 a	5.9 ± 1.4 a	6.0 ± 1.6 a	0.802

Results represent the mean value followed by the standard deviation of 61 assessments (n = 61). The attributes were assessed using a structured 9-level hedonic scale in which level 1 stands for “I do not like it at all” and level 9 stands for “I like it very much”. Those attributes marked with an asterisk were assessed on a structured 5-level hedonic scale in which level 1 stands for “much milder than I would like” or “much less sweet than I would like”, level 2 stands for “a little milder or a little less sweet than I would like”, level 3 stands for “just as I like”, level 4 stands for “a little more intense or a little sweeter than I would like” and level 5 stands for “much more intense or much sweeter than I would like”. Different letters indicate significant differences between samples according to Duncan’s test (*p* < 0.05).

## Data Availability

Data is contained within the article or Appendix A.

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
