# Peer review of "Quality and Nutritional Changes of Traditional Cupcakes in the Processing and Storage as a Result of Sunflower Oil Replacements with Refined Olive Pomace Oil"

_foods, 2023, doi:10.3390/foods12112125_

Round 1
Reviewer 1 Report
1.The abstract provides a solid overview of the study and its main findings, but could benefit from some additional details and background. For example, what is the significance of the reduction in oxidized lipid levels and preservation of nutritional properties in OPO? What impact might this have on consumers or the wider food industry?
2.The introduction provides a comprehensive overview of the background, characteristics, and potential benefits of olive pomace oil (OPO) as a substitute for sunflower oil (SO) in baked goods. The paragraph also emphasizes the limited research on the use of OPO in baked goods and the need for further investigation into its bioactive components and oxidative stability during processing and storage. One improvement suggestion is to clarify the purpose and scope of the study earlier in the paragraph. The purpose and specific baked good (cupcakes) are only mentioned in the last two sentences. It would be better to explicitly state at the beginning of the paragraph that the aim is to explore the potential use of OPO as a substitute for SO in cupcakes to improve their nutritional value and oxidative stability.
3.Result 3 mentions the use of hydroperoxide triglycerides to evaluate the degree of oxidation, but it is not clear if this method is supported by literature. It would be helpful to provide some background information or a reference to support the use of this method.
4.Result 4 states that OPO protected bioactive components such as squalene and plant sterols better than SO, but does not provide possible reasons or literature support. It is recommended to provide more explanation or literature support for these findings.
5.Overall, the study has moderate depth and innovation, but has good potential for market application and effective use of olive pomace oil. A major issue is the lack of discussion of possible causes that may have led to the research results. It is recommended to provide more discussion and explanation.
Moderate editing of English language
Author Response
Reviewer 1. Comments and Suggestions for Authors
1.The abstract provides a solid overview of the study and its main findings, but could benefit from some additional details and background. For example, what is the significance of the reduction in oxidized lipid levels and preservation of nutritional properties in OPO? What impact might this have on consumers or the wider food industry?
Including specific information related to the significance of reducing oxidised lipid levels in cupcakes or the preservation of OPO nutritional properties and the impact this might have on consumers is not an easy task given that the abstract is limited to 200 words maximum. Honestly, the abstract complies with the journal guidelines in terms that it places the research addressed in a broad context, highlights the purpose of the study, summarises the article’s main findings and indicates the main conclusions and interpretations. According to suggestions made by reviewer 2, we have added more information about the methodology used to evaluate lipid oxidation, which was based on the analysis of hydroperoxide triglycerides by HPLC (see lines 12-13), and we have also included the standard deviation found for the results presented.
It results evident that the preservation of the nutritional properties of OPO has a positive impact on consumers because they can totally benefit from its health benefits, which lies in its fatty acid composition and especially its valuable bioactive components. Recent nutritional studies have shown that regular consumption of OPO contributes to cardiovascular and cardiometabolic disease prevention. However, such beneficial effects were not found when sunflower oil or high oleic sunflower oil was used instead (see lines 48-49 in the revised version).
The reduction of oxidised lipids in cupcakes improves their quality because hydroperoxides, the most abundant oxidised lipids, eventually decompose producing volatile compounds that impart off-flavours. In addition, oxidised lipids are considered to be detrimental to health and it is recommended to decrease their levels in foods.
Although the aforementioned explanations have not been considered in the abstract section due to the length limitation, they have been included in the conclusions section for a better understanding (lines 584-588). The significance of reducing the levels of lipid oxidation in cupcakes had already been considered in the results and discussion section (see lines 484-488 in the revised version). In addition, in order to highlight the beneficial properties of OPO compared to other vegetable oils, included SO, a new sentence has been provided in the introduction section (lines 48-49).
2. The introduction provides a comprehensive overview of the background, characteristics, and potential benefits of olive pomace oil (OPO) as a substitute for sunflower oil (SO) in baked goods. The paragraph also emphasizes the limited research on the use of OPO in baked goods and the need for further investigation into its bioactive components and oxidative stability during processing and storage. One improvement suggestion is to clarify the purpose and scope of the study earlier in the paragraph. The purpose and specific baked good (cupcakes) are only mentioned in the last two sentences. It would be better to explicitly state at the beginning of the paragraph that the aim is to explore the potential use of OPO as a substitute for SO in cupcakes to improve their nutritional value and oxidative stability.
We honestly reckon that the order the information was presented in the introduction section was quite logical. OPO is not a common oil and for this reason it was presented first. Then we introduced their composition and nutritional properties followed by its great resistance to oxidative changes at high temperatures to set the starting hypothesis of this work (lines 71-74). Finally, cupcakes were presented as a study model for bakery foods. Specific cupcakes were selected in this study for several reasons, including their high market share, high fat content and high shelf life. We find this line of reasoning more appropriate than setting the goal of the study at the beginning without first supporting with references why the purpose of the research can be considered novel and timing. Furthermore, the reader can get the essential of the work in the abstract section so they will have already known what the work is about before reading the introduction.
3. Result 3 mentions the use of hydroperoxide triglycerides to evaluate the degree of oxidation, but it is not clear if this method is supported by literature. It would be helpful to provide some background information or a reference to support the use of this method.
Specific information supporting the use of the HPLC method to evaluate hydroperoxide triglycerides has been included in lines 366-371. Although the method had been quoted in the original work, we have also included an additional bibliographic reference (reference 43 in the revised version) supporting the analysis of linoleate hydroperoxide triglycerides as a good analytical approach to evaluate lipid oxidation in common oils.
4. Result 4 states that OPO protected bioactive components such as squalene and plant sterols better than SO, but does not provide possible reasons or literature support. It is recommended to provide more explanation or literature support for these findings.
Bioactive components such as squalene, alpha-tocopherol and others are minor components naturally present in oils (see lines 38-45) and they are not found in any other ingredient of cupcakes. They remain in the fat material. Therefore, it is not that OPO protected bioactive components better than SO, but the losses of bioactive components, specifically alpha-tocopherol, were lower in the OPO compared to SO because of its higher resistance to oxidative degradation. Anyway, the focus in this respect was that the OPO preserved its valuable bioactive components rather than a comparison with those in the SO. In order to make it clearer we have added an explanation on the reason why the losses of alpha-tocopherol were lower in OPO at the end of the shelf-life (lines 512-514). In addition, the high stabilities of the oil bioactive components evaluated during the shelf-life in the cupcakes have been discussed employing new bibliographic references on long-term storage studies in virgin olive oil (lines 517-532).
5.Overall, the study has moderate depth and innovation, but has good potential for market application and effective use of olive pomace oil. A major issue is the lack of discussion of possible causes that may have led to the research results. It is recommended to provide more discussion and explanation.
Accordingly, the discussion about the preservation of bioactive components in the OPO sample has been extended (line 517-532), as well as explanations on the lower losses of alpha-tocopherol found in the OPO sample have been provided (lines 512-514).
Reviewer 2 Report
The manuscript is well written and presents some interesting findings. The hypothesis is well explianed and sound. The language is clear and easy to understand.
I have following observations
i. Please maintain uniformity in the refined OPO or OPO
ii. Keyword: plz add sunflower oil also
iii. Abstract: I found first 3-4 lines very general and may be deleted; Also plz add some more details in the methodolgy in place of this. Please presents results with level of significance toimprove the impact of the findings.
iv. In introduction, please add more information on Sunflower oil in comparison to OPO specially at Para 4 for further strengthen the hypothesis.
v. Storage assay: package materials plz
vi. Please mention the assay intervals
vii. In consumer test- SO-1, OPO25-1, OPO50-1 and OPO-1? Mention it and also add footline at Table for the same
viii. In statistical analysis: please mention the sample size
ix. Table 2: SO1,2 and 3- means replicates?, please explain in footnote.
x. Results and discussion: Appropriate and well supported by suitable references
Author Response
Reviewer 2. Comments and Suggestions for Authors
The manuscript is well written and presents some interesting findings. The hypothesis is well explained and sound. The language is clear and easy to understand.
I have following observations
i. Please maintain uniformity in the refined OPO or OPO
OPO stands for olive pomace oil and it was used to refer to the oil commercialised to the public, which is a blend of refined olive pomace oil and virgin olive oil. When necessary, “refined OPO” was employed to refer to the oil used in the present study. We honestly think that no any further clarifications are required.
ii. Keyword: plz add sunflower oil also
It has been added accordingly.
iii. Abstract: I found first 3-4 lines very general and may be deleted; Also plz add some more details in the methodolgy in place of this. Please presents results with level of significance to improve the impact of the findings.
The abstract has been modified following the reviewer’s indications. Regarding the methodology, it has been indicated that the analysis of hydroperoxides was made by HPLC. The levels of significance have also been included.
iv. In introduction, please add more information on Sunflower oil in comparison to OPO specially at Para 4 for further strengthen the hypothesis.
More information on sunflower oil has been added to strengthen the starting hypothesis (lines 74-78).
v. Storage assay: package materials plz
The package material had already been indicated in the original version of the manuscript (See lines 131-133). Anyway, it has also been included in the storage assay section (line 228).
vi. Please mention the assay intervals
It has been mentioned accordingly (line 229).
vii. In consumer test- SO-1, OPO25-1, OPO50-1 and OPO-1? Mention it and also add footline at Table for the same
The samples selected for the storage assay were defined in the materials and methods section. We do not see it necessary to mention them any more in the text, nor in the footnote of Table 8. We have included the selection criterion adopted, which was random (line 227). Regarding Table 8 (Now Table 9 in the revised version), we have changed its format, from horizontal to vertical orientation, for better illustration, as well as that of tables S10-S15 in the supplementary material.
viii. In statistical analysis: please mention the sample size
The sample size has been mentioned in lines 287-290.
ix. Table 2: SO1,2 and 3- means replicates?, please explain in footnote.
As described in the materials and methods section, they refer to samples prepared with different oil batches. For better understanding, Table S1, which illustrates the oils used in the preparation of each cupcake sample, has been passed to Table 1 and the table numbers of the following ones have been changed accordingly.
x. Results and discussion: Appropriate and well supported by suitable references
We appreciate the comment.
Reviewer 3 Report
This work presents Quality and nutritional changes of traditional cupcakes in the processing and storage as a result of sunflower oil replacements with refined olive pomace oil
Title: Fine
Abstract: Fine
Keywords: Please change it to aims of the study, be specific
Introduction: Some shortcomings are below
The introduction of the article about Olive Pomace Oil (OPO) is well written and informative. However, to improve it further, authors may consider adding a brief statement that summarizes the main points of the article. This will give readers a better idea of what to expect from the article before they delve into the details. Additionally, authors may include some background information on the importance of using oils with high oxidative stability in food preparation. This will give readers a better understanding of the significance of the study and why it is important to explore the possibilities of OPO as a substitute for other oils in baked foods.
Methods and materials:
In experiment 2.1 to improve it further, authors may consider more information on the specific methods used to prepare the blends of SO and OPO, including the equipment used and the duration of the blending process. Additionally, it would be helpful to provide more information on the characteristics of the refined oils used in the study, such as their acidity levels and oxidative stability.
In experiment 2.2 some information regarding the experimental design and methodology is missing.
In experiment 2.3 Please add the purpose of this experiment may be to evaluate the quality and shelf-life of the cupcakes by assessing the moisture and fat content, oxidative stability, and color parameters.
Other parameters are OK
Results and discussion:
Please provide significant difference in table 1. Differencing difference with different alphabets
Same for table 2
Please remove the data from top of the bar chart in figure 2 and 3
Check the paper format and positions of figures in manuscript
Conclusion: Fine
Author Response
Reviewer 3. Comments and Suggestions for Authors
This work presents Quality and nutritional changes of traditional cupcakes in the processing and storage as a result of sunflower oil replacements with refined olive pomace oil
Title: Fine
Abstract: Fine
Keywords: Please change it to aims of the study, be specific
A few new keywords have been included.
Introduction: Some shortcomings are below
The introduction of the article about Olive Pomace Oil (OPO) is well written and informative. However, to improve it further, authors may consider adding a brief statement that summarizes the main points of the article. This will give readers a better idea of what to expect from the article before they delve into the details. Additionally, authors may include some background information on the importance of using oils with high oxidative stability in food preparation. This will give readers a better understanding of the significance of the study and why it is important to explore the possibilities of OPO as a substitute for other oils in baked foods.
A brief statement that summarises the main points of the article is found in the abstract and in the graphical abstract, so the reader already knows what the article is about when reading the introduction section. Honestly, we do not see it necessary to provide again a brief summary of the main aspects of the article in the introduction.
We have included a few sentences to highlight the importance of using OPO instead of SO from a nutritional point of view (lines 48-49) and also in terms of oxidative stability (lines 74-78).
Methods and materials:
In experiment 2.1 to improve it further, authors may consider more information on the specific methods used to prepare the blends of SO and OPO, including the equipment used and the duration of the blending process. Additionally, it would be helpful to provide more information on the characteristics of the refined oils used in the study, such as their acidity levels and oxidative stability.
Information about the equipment used in the oil blend preparation as well as the conditions applied has been included in the new version of the manuscript (lines 110-117). The characteristics of the oils used in the study are described in section 3.1 (Oil characterisation) and results can be found in the supplementary material, specifically in tables S1-S4, which list the results obtained for the fatty acid compositions, acidity, peroxide value and the oxidative stability index as determined by the Rancimat test at 110 ºC.
In experiment 2.2 some information regarding the experimental design and methodology is missing.
A new sentence has been included (lines 148-149) and the analysis of polar compounds has been moved to section 2.5 because it was not applied to the fresh oils (lines 204-206).
In experiment 2.3 Please add the purpose of this experiment may be to evaluate the quality and shelf-life of the cupcakes by assessing the moisture and fat content, oxidative stability, and color parameters.
Following the reviewer’s suggestions, the purpose of the cupcake characterisation has been described in section 2.3 (lines 160-161 and 187-190).
Other parameters are OK
Results and discussion:
Please provide significant difference in table 1. Differencing difference with different alphabets
Same for table 2
ANOVA and Duncan’s test for multiple mean comparisons have been applied to results in tables 1 and 2 (now Tables 2 and 3). Significant differences have been indicated using different letters. Lowercase and uppercase letters were used in Table 3 to express respectively differences between individual samples, regardless of the type of oil, and between samples prepared with different types of oil. A few remarks have been introduced in the results and discussion section to indicate the similarities and differences found between the fresh oils as a result of the statistical analysis (lines 306-308). The statistical analysis in Table 3 has been very useful and it helps understand the average values of OSI given in lines 343-344. We appreciate the reviewer’s indication very much.
Please remove the data from top of the bar chart in figure 2 and 3
They have been removed according to the reviewer’s suggestions.
Check the paper format and positions of figures in manuscript
The order of appearance of figures and tables has been checked out.
Conclusion: Fine
Round 2
Reviewer 3 Report
no further comments, accepted.